# Vision Screening Assessment (VISA) tool: diagnostic accuracy validation of a novel screening tool in detecting visual impairment among stroke survivors

Fiona J Rowe [1], Lauren Hepworth [1], Claire Howard [1], Alison Bruce [2,3] Victoria Smerdon,[4] Terry Payne,[5] Phil Jimmieson,[5] Girvan Burnside[6]

¹Health Services Research, University of Liverpool, Liverpool, UK
²Bradford Institute for Health Research, Bradford Teaching Hospitals NHS Foundation Trust, Bradford, UK
³Health Sciences, University of York, York, UK
⁴Orthoptics, Arrowe Park Hospital, Wirral, UK
⁵Computer Science, University of Liverpool, Liverpool, UK
⁶Biostatistics, University of Liverpool Faculty of Health and Life Sciences, Liverpool, UK

**Correspondence to**
Prof Fiona J Rowe;
rowef@liverpool.ac.uk

## ABSTRACT

**Purpose** Screening for visual problems in stroke survivors is not standardised. Visual problems that remain undetected or poorly identified can create unmet needs for stroke survivors. We report the validation of a new Vision Impairment Screening Assessment (VISA) tool intended for use by the stroke team to improve identification of visual impairment in stroke survivors.

**Methods** We conducted a prospective case cohort comparative study in four centres to validate the VISA tool against a specialist reference vision assessment. VISA is available in print or as an app (Medicines and Healthcare products Regulatory Agency regulatory approved); these were used equally for two groups. Both VISA and the comprehensive reference vision assessment measured case history, visual acuity, eye alignment, eye movements, visual field and visual inattention. The primary outcome measure was the presence or absence of visual impairment.

**Results** Two hundred and twenty-one stroke survivors were screened. Specialist reference vision assessment was by experienced orthoptists. Full completion of screening and reference vision assessment was achieved for 201 stroke survivors. VISA print was completed for 101 stroke survivors; VISA app was completed for 100. Sensitivity and specificity of VISA print was 97.67% and 66.67%, respectively. Overall agreement was substantial; K=0.648. Sensitivity and specificity of VISA app was 88.31% and 86.96%, respectively. Overall agreement was substantial; K=0.690. Lowest agreement was found for screening of eye movement and near visual acuity.

**Conclusions** This validation study indicates acceptability of VISA for screening of potential visual impairment in stroke survivors. Sensitivity and specificity were high indicating the accuracy of this screening tool. VISA is available in print or as an app allowing versatile uptake across multiple stroke settings.

## INTRODUCTION

The prevalence of overall visual impairment has been estimated at 65%–73% with varying prevalence reported for specific types of visual impairment (inclusive of reduced central vision, ocular motility defects, visual field loss and visual perception problems).[1–4]

### Strengths and limitations of this study

► Validation of the Vision Impairment Screening Assessment (VISA) screening tool in this prospective study shows improved detection accuracy for detection of stroke-related visual impairment.
► The study included clinicians involved in stroke care who are not specialists in vision problems and lack formal eye training.
► Where early visual impairment detection occurs, this facilitates prompt referral with fewer false positives and negatives.
► Through process evaluation, clinicians reported acceptability of the VISA screening tool for is use in screening for presence of vision problems in stroke survivors.
► The VISA screening tool may further be of potential use for visual screening in other care settings such as neurorehabilitation.

Figures for the incidence of new-onset visual impairment following stroke are placed at about 60%.[4] Given the estimated 100 000 new-onset strokes per annum in the UK there are sizeable numbers of stroke survivors living with stroke-related visual impairment.[5]

Visual impairment constitutes a considerable comorbidity of stroke. Visual impairment, on its own or in addition to other stroke-related disabilities, can cause significant impact to quality of life.[6] For many, it results in inability or altered ability to undertake many aspects of daily activities with impact on return to work, participation in hobbies and family life, and can lead to social isolation, altered mood and depression.[7–9] Interventions for stroke-related visual impairment are well established[10] but require referral to appropriate eye care services, which is facilitated through orthoptic service routes.[11] Where visual impairment is identified, this facilitates optimisation of other therapy and early access to vision rehabilitation.

There are issues with how best to identify the presence of visual impairment through stroke team vision screening and specialist vision assessment.[12] Even with screening measures in place there are also issues reported with provision of care and access to vision services for stroke survivors who have been identified as having vision problems.[13]

The overall aim of this study was to validate the Vision Impairment Screening Assessment (VISA) tool which uses simple established assessments of visual function coupled with detailed instructions. Our objectives were to test VISA, available in print or as a software application, against a reference of a specialist vision assessment to determine sensitivity, specificity, predictive values and inter-rater agreement of results between VISA and specialist vision assessments.

## METHODS

The development and pilot validation of VISA have been described elsewhere.[14] This study is reported in accordance with the STandards for Reporting of Diagnostic accuracy (STARD) guidelines.[15]

### Design

A prospective case cohort comparative design was used for the validation clinical study between September 2016 and February 2019. Individuals were suitable for inclusion if they were 18 years of age or older, had clinical diagnosis of stroke as defined by WHO, had the ability to agree to vision screening using verbal or non-verbal indications of agreement, did not have severe cognitive impairment preventing screening defined as difficulty with memory/concentration/decision making and thus being unable to follow instructions, and did not decline vision screening. This was a convenience sample of participants who were identified as being eligible from inpatients on the acute stroke unit. With recruitment on the acute stroke unit, time to VISA assessment was typically within 1 week of stroke onset. Our inclusion criteria were intended to be pragmatic and inclusive of as many stroke survivors as possible. All participants provided informed consent.

### Setting, recruitment and assessment

Recruitment took place across five hospitals (secondary hospital care) in which an orthoptist was a member of the core acute stroke unit multidisciplinary team (MDT) (as per national guidelines: Royal College of Physicians Intercollegiate Stroke Guidelines and British & Irish Orthoptic Society extended guidelines for stroke practice).[16 17]

For the purpose of this study, vision screening was undertaken with VISA and screening was defined as the assessment of stroke survivors for the presence of reduced visual function against preset abnormality criteria, outlined in the statistical methodology section.

Specialist visual assessment was defined as the vision assessment undertaken by an orthoptist in which detection of visual impairment was coupled with formal diagnosis of the type of visual condition present. As a minimum this consisted of near and distance LogMAR (Logarithm of the Minimum Angle of Resolution) visual acuity, cover test, ocular motility assessment, standardised visual field to confrontation using 10 mm red targets and visual inattention assessment.

Each stroke survivor underwent two vision assessments: the routine orthoptic specialist vision assessment and the VISA screening assessment. Patients were recruited consecutively as being identified to meet the inclusion criteria and providing consent to participate.

VISA was available in print and as a software app. VISA was used in print form for the first half of recruitment and, subsequently in app form for the second half of recruitment to this study. Both VISA formats consisted of five VISA sections comprising case history, LogMAR visual acuity at near and distance, eye alignment and movement, visual fields and visual inattention. A separate section comprising stand-alone user instructions is included. In brief, VISA consists of five sections. Section 1 comprises a case history with questions and observations of visual symptoms and signs. When it is not possible to obtain a case history from the patient, the tool advises to consult family members/carers. The person completing the screen is instructed to observe for abnormalities of lids, pupils and head position among other vision signs. Section 2 comprises an assessment of LogMAR visual acuity for near (35 cm) and distance (3 m); monocular or binocular depending on the ability of the patient. A matching card was available for patients who were unable to name letters but could point to letters. For those unable to comply with any letter test, a further option included grating cards that use a preferential looking technique which is particularly useful with cognitive/communication issues. Section 3 is an assessment of eye alignment observing symmetry of the corneal reflections of each eye. Clinician observations can be compared with images of straight eyes or images of eyes in converged, diverged, elevated or depressed strabismus positions. Eye movements (smooth pursuits) assessed full movements of each eye into up, down, right and left gaze positions. Clinician observations could be compared with images of full ocular rotations to right/left gaze, elevation/depression and on convergence. Section 4 is an assessment of visual field, and section 5 is an assessment of visual inattention including line bisection, clock drawing and a cancellation task. The print and app versions are identical with the exception of the visual field assessment. In VISA print, a standardised method of confrontation is conducted. Confrontation follows a typical method with the clinician seated directly opposite the patient at a distance of 1 m and following stages that involve the patient indicating when a 10 mm red target is seen in the periphery of their vision, finger counting in each quadrant of the visual field and comparison of examiner facial features. In VISA app, a kinetic visual field assessment is undertaken which, at a test distance of 30 cm and a screen width of 24.6 cm, allows an assessment of the 40° visual field. The patient is

asked to fixate a static fixation point in the corner of the screen while a stimulus moves from the other edges. They are asked to tap the tablet screen when the stimulus is seen. This is repeated with the fixation target positioned at all four corners of the screen.

Section 5 includes three routine assessments for visual inattention; line bisection, clock drawing and a cancellation task. The line bisection task requires the patient to indicate the centre of line for three lines of differing lengths. The cancellation task requires the patient to cross out large clock symbols among distractors of small clock symbols and large/small open circles. Clock drawing requires the patient to draw the numbers and clock hands on a blank circle. VISA app collates data from each of the sections to create a PDF record of the assessment. The free-to-access VISA tool is available on; www. vision-research.co.uk.

The routine orthoptic vision assessment comprised detailed diagnostic assessments of case history, visual acuity, ocular alignment and movement, visual field and visual perception. This assessment was undertaken within 24 hours (typically the same day) of the VISA screen—to minimise effect of potential recovery. The orthoptic assessment covered all assessment sections included in the VISA tool. However, the orthoptist undertook a detailed assessment using their specialist expertise to interpret the results and adapt testing methods to individual requirements.

The order of the VISA screening and orthoptic vision assessments varied in a pragmatic manner to avoid the effects of fatigue and bias towards either the screen or orthoptic vision assessment. The screener and orthoptist were blinded to each other's assessments to prevent bias of assessment. The within-assessment order of testing varied for the orthoptic assessment. However, the order of testing within the VISA screen followed a set order of (1) case history, (2) visual acuity, (3) eye alignment and movement, (4) visual field and (5) visual inattention assessments.

### Statistical methodology and sample size

Results were taken in numerical format from the referral forms completed by both the screener and orthoptist. The orthoptic vision assessment was taken as the reference standard.

The primary outcome measure was a binary measure of the presence or absence of visual impairment (defined as one or more of the following; reduced distance vision <0.2, reduced near vision <0.3 (equivalent to N6), deviated eye position, eye movement abnormality (incomplete eye rotations in any position of gaze), visual field loss (eg, presence of hemianopia, quadrantanopia, constriction), visual inattention with displaced line bisection, <42 score on cancellation task and/or incomplete/displaced clock drawing). The primary outcome measure was evaluated by kappa values assessing chance-eliminated agreement between the results of the VISA screening and orthoptic vision assessment.

Secondary outcome measures were the calculation of sensitivity, specificity and predictive values. Level of sensitivity was estimated as the proportion of patients with visual impairment as diagnosed by the gold-standard clinical examination, which are correctly identified by the screener, and the corresponding 95% CI was calculated. Level of specificity was estimated as the proportion of patients without visual impairment that are correctly identified by the screener, and the corresponding 95% CI. Further, we calculated the positive and negative predictive values for the VISA screen. Kappa (K) values assessed chance-eliminated agreement between the individual components of VISA tool and orthoptic vision assessment. The interpretation used was 0.0–0.2 poor, 0.21–0.4 fair, 0.41–0.6 moderate, 0.61–0.8 substantial and 0.81–1.0 almost perfect.[18] Analysis was conducted using StatsDirect software (StatsDirect).

For sample size, we applied the principles for diagnostic accuracy studies, and aimed to recruit a sample of 100 for validation of VISA print and a further sample of 100 for VISA app.[19]

### Process evaluation

Process evaluation for acceptability of VISA during the clinical study was collected via clinician feedback sheets and one-to-one reports from patients. Feedback sheets could be returned at any time during the study to report any issues with testing alongside obtaining clinician views based on their use of VISA. Feedback sheets asked the following:

1. Are the instructions for the various tests clear?
2. Which instructions should be amended?
3. What additional instruction information/rewording do you suggest?
4. Which instructions require less information?
5. Are any tests not useful or difficulty to do? (Specify)
6. Should any other tests be added in?
7. How long does it take you to do the screen?
8. Other comments?

Comments collected from feedback sheets and reports were collated descriptively.

### Patient and public involvement

Patients were involved in the design and monitoring of this study. Patients from the VISable stroke and vision panel were consulted when devising the study plan and conduct. Reports during the conduct of this study were circulated to the VISable panel for patient monitoring purposes.

## RESULTS
### Completion rate

Two hundred and twenty-one stroke patients received both a VISA screening assessment and a reference vision assessment (during the period of September 2016 to February 2019).

All elements of the VISA screen were attempted by 201 patients. VISA print was used with 121 patients from which complete data was available for 101 for analysis. The mean age of patients on stroke admission was 70.6 (SD 13.5), 46 were females and 54 males. The reported mean time of test duration was 23.5 min (SD 10.0).

VISA app was completed with 100 patients with a mean age of 63.4 (SD 13.4), of which 72 were males and 28 were females.

VISA print was fully completed by 91 patients, with the remaining 10 missing one or more elements (near vision n=5, distance vision n=5, ocular motility n=1, visual fields n=1, visual inattention n=9). The orthoptic vision assessment was fully completed by 90 patients, with the remaining 11 missing one or more elements (near vision n=8, visual inattention n=9). Reasons for inability to complete one or more of the elements were typically recorded as either cognitive impairment or fatigue. VISA app and orthoptic vision assessment were fully completed by all 100 patients. Missing data did not automatically result in failure for that section, thereby requiring referral. The reason for failure was taken into account; for example, if a section was not completed due to fatigue this would not pragmatically have resulted in a referral but instead, a retest.

### Referral agreement for VISA print

The agreement of whether to make a referral to specialist eye services based on the results of the VISA print versus those from orthoptic vision assessment had a kappa value of 0.648 (substantial agreement) (95% CI 0.424 to 0.872).

Sensitivity of 97.67% and specificity of 60.00% were found. The positive and negative predictive values were 93.33% and 81.82%, respectively. These calculations are outlined in table 1. Agreement was found for 93 participants (9 had no visual impairment, 84 required referral because of failed screening) as outlined in figure 1.

VISA print produced two false negative and six false positive results. Of the false negative results, both had ocular motility problems, of which one also had reduced near vision. The two ocular motility problems missed were asymptomatic minimal rotary nystagmus and limited elevation. The latter also had reduced near vision at 0.450 LogMAR. For false positive results, three with reduced near vision, two with ocular motility problems and one with both reduced near vision and visual inattention, were detected by screening and found not to be present by the orthoptic vision assessment. The referrals relating to reduced near vision all detected N8 level of vision in one or both eyes. The referral relating to visual inattention was detected on the clock drawing element; it was noted by the examiner that the inaccurate completion was likely due to cognitive impairment. The ocular motility problems detected were reported as limitation of vertical gaze and nystagmus.

**Table 1** Calculations of sensitivity, specificity and predictive values for VISA print

| Positive, that is, pathologic n=86 | |
|---|---|
| True positive, that is, visual impairment present and referred | 84 |
| False negative, that is, visual impairment present but not referred | 2 |
| **Negative that is, normal n=15** | |
| False positive, that is, visual impairment not present but referred | 6 |
| True negative, that is, visual impairment not present and not referred | 9 |
| **Output** | |
| Sensitivity (true positive/ true positive+false negative) | 97.67% (95% CI 91.85% to 99.72%) |
| Specificity (true negative/ false positive+true negative) | 60.00% (95% CI 32.29% to 83.66%) |
| Positive predictive value (true positive/false positive+true positive) | 93.33% (95% CI 88.27% to 96.30%) |
| Negative predictive value (true negative/false negative+true negative) | 81.82% (95% CI 51.83% to 94.95%) |

VISA, Vision Impairment Screening Assessment.

### Test component agreement for VISA print

The agreements for the individual components between VISA print and orthoptic vision assessments are outlined in table 2. The highest levels of agreement were produced for distance visual acuity (0.565) and visual fields (0.504), both with moderate agreement. The lowest level of agreement was produced for near visual acuity (0.236) and ocular motility (0.367), both with fair agreement. Low agreement for ocular motility related to high false positives and false negatives. Ten cases (one with multiple conditions) were not detected (false negative). These comprised four defects of vertical movement (including one upgaze palsy, two restrictions of elevation and one V-pattern), three cases of nystagmus (including one minimal rotary nystagmus, one gaze-evoked and one end-point nystagmus) and four cases of reduced convergence. The low agreement with near visual acuity related to high false negatives where 23 cases were not detected—these composed of 10 with 0.4 LogMAR or better, nine between 0.4 and 0.5 LogMAR and three 0.6 LogMAR or worse.

### Referral agreement for VISA app

The agreement of whether to make a referral to specialist eye services based on the results of VISA app versus those

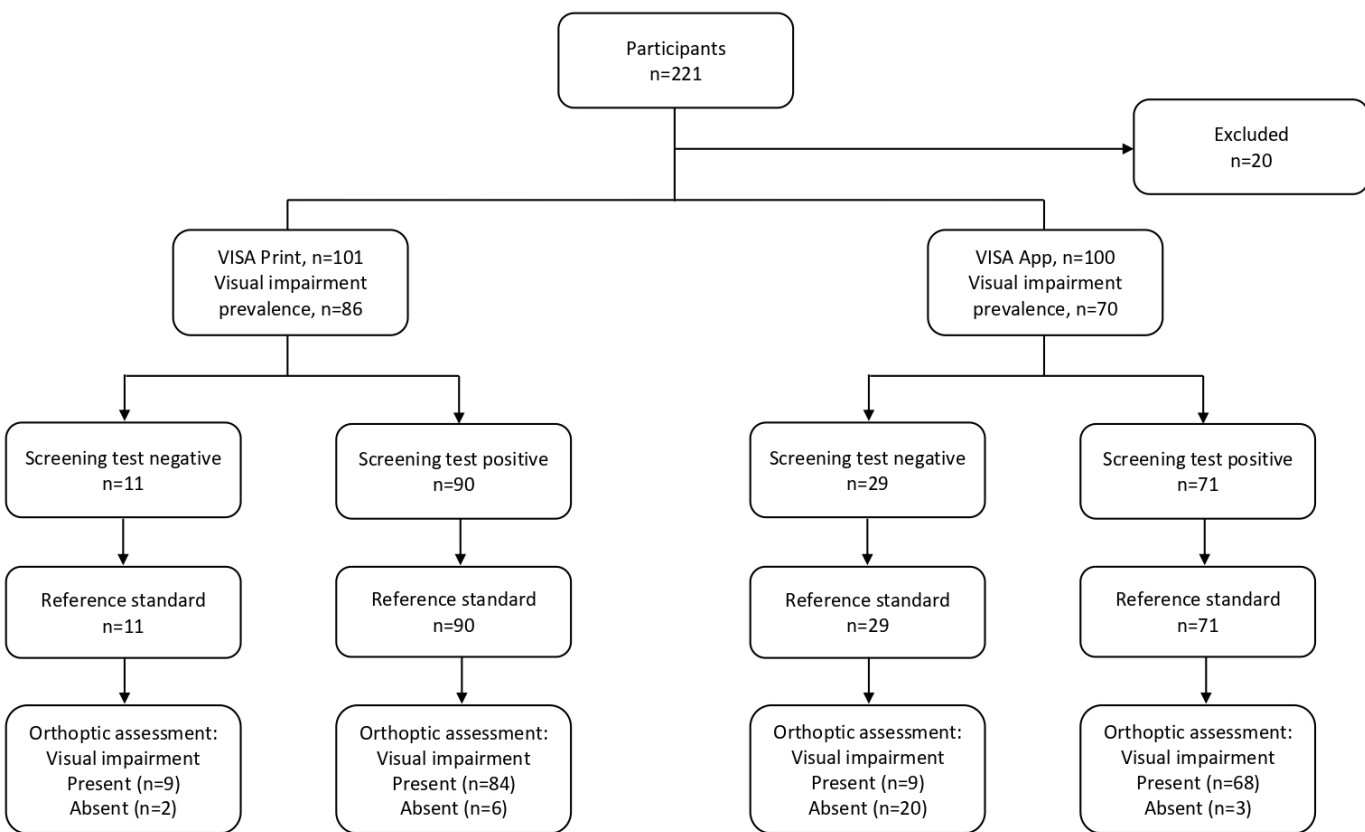

**Figure 1** Flow diagram of participant outcome for VISA screening and orthoptic full assessment. VISA, Vision Impairment Screening Assessment.

from orthoptic vision assessment had a Kappa value of 0.690 (substantial agreement) (95% CI 0.528 to 0.851).

Sensitivity of 88.31% and specificity of 86.96% were calculated. The positive and negative predictive values were 95.77% and 68.97%, respectively. These calculations are outlined in table 3.

Agreement was found for 88 participants (20 had no visual impairment, 68 required referral because of failed screening) as outlined in figure 1. VISA app produced nine false negative and three false positive results. Of the false negative results, four had slightly reduced near vision between 0.3 and 0.4 LogMAR, two had reduced distance vision of 0.3 LogMAR, two had mild visual inattention

(one detected on clock cancellation and was the examiners judgement) and one had reduced near vision of 0.4 LogMAR and a visual field defect (partial right superior quadrantanopia detect on confrontation only, but not detected by formal perimetry using binocular Esterman). False positive results (one with reduced distance vision, one with a visual field defect and one with both a visual field defect and visual inattention) were detected by screening and found not to be present by the orthoptic vision assessment. The referral relating to reduced distance vision was 0.4 LogMAR. The referral relating to a visual field defect of general constriction and visual inattention was detected on the longest line in the line

**Table 2** Summary of agreement between VISA print and orthoptic vision assessment for referral to specialist eye services and individual components

| Element of testing | Agreement | False negative | False positive | Kappa value (95% CI) |
|---|---|---|---|---|
| Referral | 93 | 2 | 6 | 0.648 (0.424 to 0.872) |
| Near visual acuity | 65 | 23 | 12 | 0.236 (0.045 to 0.427) |
| Distance visual acuity | 79 | 9 | 13 | 0.565 (0.405 to 0.725) |
| Ocular alignment | 89 | 5 | 7 | 0.388 (0.110 to 0.667) |
| Ocular motility | 72 | 10 | 19 | 0.367 (0.181 to 0.553) |
| Visual fields | 76 | 7 | 18 | 0.504 (0.339 to 0.668) |
| Visual inattention | 74 | 4 | 21 | 0.500 (0.340 to 0.659) |

VISA, Vision Impairment Screening Assessment.

**Table 3** Calculations of sensitivity, specificity and predictive values for VISA app

| Positive, that is, pathologic n=77 | |
|---|---|
| True positive, that is, visual impairment present and referred | 68 |
| False negative, that is, visual impairment present but not referred | 9 |
| **Negative that is, normal n=23** | |
| False positive, that is, visual impairment not present but referred | 3 |
| True negative, that is, visual impairment not present and not referred | 20 |
| **Output** | |
| Sensitivity (true positive/true positive+false negative) | 88.31% (95% CI 78.97% to 94.51%) |
| Specificity (true negative/false positive+true negative) | 86.96% (95% CI 66.41% to 97.22%) |
| Positive predictive value (true positive/false positive+true positive) | 95.77% (95% CI 88.72% to 98.49%) |
| Negative predictive value (true negative/false negative+true negative) | 68.97% (95% CI 54.10% to 80.73%) |

VISA, Vision Impairment Screening Assessment.

bisection element. The other visual field defect detected was general constriction.

### Test component agreement for VISA app

The agreements for the individual components between VISA app and orthoptic vision assessments are outlined in table 4. The highest levels of agreement were produced for distance visual acuity (0.783) and visual fields (0.701), both with substantial agreement. The lowest level of agreement was produced for visual inattention (0.323) with fair agreement. The low agreement with visual inattention related to 16 false positives, of which 13 were detected with one of the three tests; 12 with line bisection, 1 with clock drawing.

### Perimetry agreement

Twenty-five participants had formal perimetry using the binocular Esterman programme rather than confrontation. There was perfect agreement (1.0) of whether a visual field defect was present between the kinetic visual field test on VISA app versus formal perimetry using the binocular Esterman programme. Twenty-one had a visual field defect and four were found to have a normal visual field.

### Process evaluation

Information from feedback sheets and detailed notes from interviews were compiled and grouped for type of feedback. Minimal feedback was obtained during the validation study. Feedback related to the duration of screening, presentation of tests on the app and referral guides. One stroke unit noted that VISA could take too long in the hyperacute stage with unwell patients. Feedback on app presentation included a change to the clock drawing circle (to remove lines that might indicate time markers), change to the fixation target for the visual field test, addition of a nystagmus check on eye movement testing (in addition to its presence in the case history checklist) and ability to delete erroneous marks on the line bisection test. For referral guidance, feedback requested the addition of a refer/retest icon on the patient results page. Further feedback reported greater ease of screening with the app for those having to use their non-dominant hand because of upper limb motor impairment. Stroke survivors found it easier to respond using the touch screen than traditional pen and paper tasks when using their non-dominant hand.

### DISCUSSION

In this study, we present the VISA screening tool, performed by non-eye trained specialists, with validation results for the printed version and for the software app. Overall, referral had sensitivity and specificity of >88% and >60%, respectively, positive and negative predictive values of >93% and >68%, respectively, with substantial agreement between VISA screening and comprehensive orthoptic assessment of

**Table 4** Summary of agreement between VISA app and orthoptic vision assessment for referral to specialist eye services and individual components

| Element of testing | Agreement | False negative | False positive | Kappa value (95% CI) |
|---|---|---|---|---|
| Referral | 88 | 9 | 3 | 0.690 (0.528 to 0.851) |
| Near visual acuity | 77 | 19 | 3 | 0.416 (0.227 to 0.605) |
| Distance visual acuity | 90 | 6 | 4 | 0.783 (0.656 to 0.910) |
| Visual fields | 85 | 3 | 12 | 0.701 (0.564 to 0.838) |
| Visual inattention | 78 | 6 | 16 | 0.323 (0.108 to 0.538) |

VISA, Vision Impairment Screening Assessment.

about kappa 0.7. Agreement was lowest for eye movement screening, near visual acuity and visual inattention whereas all other individual sections showed higher levels of agreement. Process evaluation aided further refinement of VISA and, in particular, changes to presentation features on the app version.

When designing and using screening tools there is a balance between sensitivity and specificity for reliable detection of deficits. Low agreement in the VISA sections related to high false positive referrals where VISA screen indicated a fail for ocular motility or visual inattention. The orthoptic vision assessment confirmed ocular motility changes which were classed as 'normal' physiological eye movement patterns such age-related reduced elevation, and which alone would not have required referral. False positive referrals for visual inattention occurred where the patient failed to complete the section because of fatigue or cognitive impairment. Reduced visual acuity was always at a borderline level just above the fail threshold.

False negative referrals are important to consider; failed detection of significant deficits is to be avoided. Our results showed low numbers of false negatives which included failed detection of ocular motility defects, reduced visual acuity, visual inattention and visual field defect. The ocular motility defects were related to asymptomatic limited elevation and minimal nystagmus which would not have constituted referrals by orthoptic vision assessment. Reduced visual acuity, similar to the false positive results, was always close to the pass/fail threshold. Arguably, this is an ideal call for retest rather than refer. One case of mild visual inattention was not passed by VISA app where the diagnosis had been made by clinical observation. One visual field defect related to a peripheral field loss; a defect that could not be detected by the central testing area of VISA app.

Specificity was higher when using VISA app compared with VISA print. Mean age for the VISA app group was lower than the VISA print group with more male than female participants in the VISA app group. It is unlikely that age/sex differences affected agreement between the VISA print/app formats versus orthoptic assessment as all participants, by default of meeting the inclusion criteria, were able to undergo both assessments. Differences are more likely due to the staff mix using VISA. VISA print was used solely by members of the stroke team and often without any formal vision training. VISA app was used by a mix of stroke team members but also orthoptists. Accuracy was likely enhanced by involvement of the latter. Referral agreements overall for decision on making a referral were 0.648 (VISA print) and 0.690 (VISA app), both indicating substantial agreement. It should be noted that kappa is dependent on the base rate of the outcome being assessed, with calculated values being lower when the prevalence of the outcome is either very high or very low. Bruckner and Yoder suggest estimating overall accuracy using a combination of kappa and base rate of outcome.[20] This method does not change the conclusions drawn here, as these outcomes with substantial agreement have estimated accuracy of at least 90% when base rate is taken into account,

and those with moderate agreement at least 85% estimated accuracy.

VISA print and app provide a vision screen across the main categories of potential visual impairment following stroke. Besides a case history section, screening includes visual acuity, eye position and movements, visual fields and visual inattention. There are potential advantages for using either the manual tool or the app. Some clinicians and stroke survivors may prefer and respond better to use of traditional testing options inclusive of pen and paper tasks. The recording charts are completed during the testing period and can be entered in hospital case notes immediately. The app produces a PDF file of results which has to be printed before entry in hospital case notes. Conversely, the PDF file is an advantage for electronic hospital records. Further, the app provides a constant background illumination for screening assessments whereas the manual is used under variable lighting conditions dependent on wherever the screening is undertaken. The app uses a kinetic central visual field assessment that is run as a standardised test which reduces examiner bias—a bias that persists for confrontation visual field assessment.

When developing and validating a screening tool it is important that it is compared with a gold standard. In our UK services, the gold standard is an orthoptic assessment undertaken on the acute stroke unit. The development and pilot of VISA followed a robust process.[14] In this follow-on validation study we considered the results of VISA versus the gold-standard orthoptic assessment in evaluating construct and content validity. We further considered ecological validity through use of the tool by clinical (not research) stroke teams in the real-world environment of busy acute stroke units in the UK National Health Service (NHS). Additionally, we sought specific feedback through process evaluation collecting feedback forms from stroke team clinicians and patient reports.

A systematic review of screening options for poststroke visual impairment reported vision screening checklists and stroke screening tools (eg, National Institute for Health Stroke Scale (NIHSS), Face Arm Speech Test - Ataxia, Visual field defect, Vertigo, Vomiting (FAST-AVVV), Ataxia, Blindness, Consciousness, Dysphagia, Eye 1 (diplopia), Eye 2 (pupils) (ABCD-E2)) which include elements of vision assessment, however, not all potential visual impairments are screened.[21] Past vision screening publications have reported the results of vision 'checklists'—lists of information gathered from questioning the patient, observations or from data documented in the case notes. The Vision In Stroke (VIS) study reported checklist screening in 915 stroke survivors with sensitivity of 0.42, specificity of 0.52 and agreement against a reference standard of 0.428 (95% CI –0.048 to 0.019: kappa).[2 12] An Australian study reported the use of a checklist for detection of eye conditions and vision defects in 100 stroke survivors with 69% accuracy and intraclass correlation of 0.84 (95% CI 0.77 to 0.89).[22] More recently, a vision screening app (available on android platforms) was developed for use with stroke survivors (StrokeVision app) with sections assessing visual acuity,

visual fields and visual inattention.[23] This was validated with a cohort of 45 stroke survivors with sensitivities across the various sections of 50%–79% and specificities of 87%–98%. The specificities reported are higher than those from our VISA study but likely reflect the use of the StrokeVision app by fully trained research assistants vs the VISA completion by members of the stroke MDT who only followed the in-built screening instructions.

Overall, vision tools/apps provide a more extensive screening of vision with greater accuracy than vision checklists. However, there are issues with how best to identify the presence of visual impairment through stroke team vision screening and specialist vision assessment.[21 24] Even with screening measures in place, there are also issues reported with provision of care and access to vision services for stroke survivors who have been identified as having vision problems.[13]

An ideal stroke vision service follows recommendations from the National Clinical Guidelines for Stroke which specify orthoptists as core members of the acute stroke team and screen all stroke survivors prior to discharge.[16] Despite consistent findings that inclusion of vision services within the MDT is highly beneficial, such visual assessment is not common and services are inconsistent throughout the UK. Stepped models of care must be considered to meet the needs of stroke survivors against the context of local service capacity. Access to orthoptic services on acute stroke units enables faster provision of vision screening. The earlier assessment time point reported for the IVIS study is important as it shows the feasibility and acceptability of early visual assessment within 3 days of stroke onset for at least half of stroke survivors and within 1 week of stroke onset for the majority.[4] Early detection of visual impairment is important. Although some cases of visual impairment will recover quickly, the majority do not. Moreover, there are few predictive factors for who will recover.[4] This prompt early detection, in turn, allows early detection of visual impairment and sharing of the functional significance of this with the patients, carers and stroke teams. Furthermore, early assessment leads to early intervention which has potential impact on general rehabilitation where visual function can be improved.[2 3 10] In the absence of orthoptic services, further stepped down models of care include the use of screening tools or screening checklists.

Such screening methods cannot replace the accuracy of a reference, specialist vision assessment. However, they serve an important purpose of obtaining a standardised screen in the absence of on-site specialist vision services and are better than no or non-standardised assessments by non-eye trained clinicians. In such instances, we advise the use of a screening tool. The advantages of VISA print and app are their validation in a real-world pragmatic study conducted in acute stroke units and used by non-eye trained clinicians. Clinicians used the in-built standalone instructions— designed to avoid the need of regular specialist vision training which can be difficult to access or provide. Further, the app is MHRA approved for clinical assessment; an important requirement for NHS adoption. The availability of VISA as a manual and as an app facilitates use alongside paper-based records or integration with electronic patient record systems.

Vision checklists have been shown to have a low sensitivity and specificity, and an over-reliance on the report of visual symptoms.[2 12] The VISA print and app offer an intermediate measure between vision checklists and orthoptic specialist vision services with greater accuracy than vision checklists but lacking the accuracy of orthoptic assessments and the immediate access to management of visual problems provided by orthoptic stroke unit services. The VISA print/app does not preclude the use of vision checklists, however, for some stroke survivors who are very unwell acutely and/or lack sufficient cognition and communication. Simpler vision checklists are quick and easy to use in such circumstances and remain more accurate than no vision screen at all.

There are some limitations to consider for this study. The VISA screening tool was used on acute stroke units. There was no validation of the tool in community settings. However, there is little reason to think it would be of less use or accuracy when used in other stroke settings. Test–retest and inter-rater variability of the VISA screening tool were not evaluated during this study as this would have caused too high a burden of assessment on participants. Information on education level, stroke type, stroke severity and ocular history were not obtained for this study. These sources of information were not considered essential to this study as the primary aim was to determine if VISA could detect visual impairment regardless of patient/stroke demographics and regardless of whether visual impairment was new or pre-existent. These aspects would provide potentially useful discussion in a future implementation study of VISA. A further limitation is that we included a convenience sample of stroke survivors in this study. The study was designed as a pragmatic clinical study to fit in with daily clinical practice and with minimal disruption to service and care on the acute stroke unit. As a result the stroke team were potentially more likely to screen stroke survivors at risk for visual impairment. This may explain the higher prevalence rate of visual impairment for this study (85% VISA print and 77% VISA app) than that reported in a recent epidemiology study (73%).[4]

## CONCLUSIONS

Validation of the VISA screening tool in either print or app format shows improved detection accuracy for detection of stroke-related visual impairment by clinicians involved in stroke care who are not specialists in vision problems and lack formal eye training. Where early visual impairment detection occurs, this facilitates prompt referral with fewer false positives and negatives. Clinicians reported acceptability of the VISA screening tool for is use in screening for presence of vision problems in stroke survivors. Referral sensitivity of >88% and specificity of >60% were found for the VISA screening with substantial inter-rater agreement for referral between VISA screening and specialist

vision assessments. The VISA screening tool provides a standardised and validated method to screen for visual problems following stroke and may further be of potential use for visual screening in other care settings such as neurorehabilitation.

**Acknowledgements** We thank the VISable panel for their involvement in the design and monitoring of this study. We thank the patients and staff at Arrowe Park Hospital, Bradford Teaching Hospital, Manchester University Hospital, Salford Royal Hospital and the Walton Centre; specifically Felicity Hale, Jennifer Lawrence, Sam Campbell, Claire Fowler, Imtiaz Nazir, Laura Stubbs, Abigail Kerr, Stephanie Wolff, Kim Cochrane, Tamsin Steward, Ella Henry, Carmel Noonan and Hayley Draper.

**Contributors** FJR provided oversight for the study and led the writing of the paper. FJR, LH, CH, VS and AB contributed to data collection, reviewing the draft paper and approving the final version. TP and PJ contributed to the development of VISA app software and its testing, and reviewed the draft paper and approved the final version. GB contributed to the statistical analysis and interpretation, and reviewed draft amendments and approved the final version.

**Funding** This study was supported by the University of Liverpool.

**Competing interests** None declared.

**Patient and public involvement** Patients and/or the public were involved in the design, or conduct, or reporting, or dissemination plans of this research. Refer to the Methods section for further details.

**Patient consent for publication** Not required.

**Ethics approval** The clinical study was undertaken in accordance with the Tenets of Helsinki with NHS research ethical approval. Research ethics approval was obtained separately for VISA print (16/NI/0125) and for VISA app (17/WA/0411). The app was approved by the Medicines and Healthcare products Regulatory Authority (MHRA: Reference CI/2017/0065) for NHS use in this study.

**Provenance and peer review** Not commissioned; externally peer reviewed.

**Data availability statement** Data are available on reasonable request. Data can be accessed via direct contact with the lead author.

**ORCID iDs**
Fiona J Rowe http://orcid.org/0000-0001-9210-9131
Lauren Hepworth http://orcid.org/0000-0001-8542-9815
Claire Howard http://orcid.org/0000-0002-2806-9144
Alison Bruce http://orcid.org/0000-0002-6028-6587

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
