## [Reviewer comments · BMJ Open]

ARTICLE DETAILS

TITLE (PROVISIONAL)	The Vision Screening Assessment (VISA) tool - diagnostic accuracy validation of a novel screening tool in detecting visual impairment among stroke survivors.
AUTHORS	Rowe, Fiona; Hepworth, Lauren; Howard, Claire; Bruce, Alison; Smerdon, Victoria; Payne, Terry; Jimmieson, Phil; Burnside, Girvan

VERSION 1 - REVIEW

REVIEWER	Thomas Meinel Inselspital, Bern
REVIEW RETURNED	30-Sep-2019

GENERAL COMMENTS	I read with interest the paper entitled, "The Vision Screening Assessment (VISA) tool - diagnostic accuracy validation of a novel screening tool in detecting visual impairment among stroke survivors.". The authors report prospectively collected data to report on the sensitivity and specificity of a novel standardized screening tool for visual impairment among stroke survivors (measured during acute stroke unit care, composite endpoint of reduced distance vision, reduced near vision, eye movement abnormality, visual field loss, and visual perceptual abnormality). The authors' rationale is that currently visual impairments are often neglected and current screening tools (checklists) lack sens and spec to rule in or rule out visual impairment. They conclude that their novel tool might be a diagnostic option in identifying visual impairment in stroke survivors. I like to see a diagnostic accuracy study published, however this report has several points, that need to be addressed in order to draw the correct conclusions. Inclusion criteria: Severe cognitive impairment: defined as? When was this done? Acute Stroke Care? Please report days after stroke as delirium, infections, blood pressure, ... can cause visual impairment. Discussion: "...3 days of stroke onset for at least half of stroke survivors and within 1 week of stroke onset for the majority" Suggest to do this test 4 weeks after stroke. Many motor symptoms of stroke are improving rapidly in the first days, so you might have picked up transient visual dysfunction, which might not have been relevant to the patient any more when he sees the specialist. Convenience sample: How was randomization of which patients to screen exactly done? Are you sure, you did not exclude "hard to diagnose" patients, hence boosting the sens and spec of your tool? Introduction: when 65% of all patients have visual impairment and
--

	60% of stroke survivors, then why is the visual impairment stroke-related? Misleading sentence. “For the purpose of this study, vision screening was undertaken with VISA and screening was defined as the assessment of stroke survivors for the presence of reduced visual function against pre-set abnormality criteria.” Which were they? Did you publish your protocol? “Visual field to confrontation”: Goldberg perimetry or finger? “The order of the VISA screening and orthoptic vision assessments varied”: by random? Screener and othoptist blinded to Imaging / Clinical Stroke Information? Which patients VISA app, which VISA print? Randomized? 122 VISA print patients – 101 available = 21 missings: baseline differences? Hard to diagnose? agreement of whether to make a referral to specialist: was there a predefined cutoff or any visual impairment would lead to referral? “The lowest level of agreement was produced for near visual acuity (0.236) and ocular motility (0.367), both with fair agreement.” Kappa <0.4 is usually considered poor agreement. https://www.sciencedirect.com/science/article/pii/S1556086415318876 How were the abnormalities stroke related? Please report on stroke location (occipital cortex, brainstem, cerebellum) in order to get a feeling, how much of the visual impairment was actually stroke-related. Also suggest to do this screening in a matched population without stroke. I doubt a bit, how much of the pathologies were actually stroke-related. Specificity of 60%, imagine, what this would mean when VISA would be applied to all stroke patients in terms of logistics and finance. A referral of millions of patients to eye specialists for no reason. Please discuss this impact in your discussion. Table 3: please report frequencies of pathologic and normal results, not only agreement. For which of these pathologies, there is actually an evidence-based treatment?
--	--

REVIEWER	Giovanni Galeoto Sapienza University of Rome
REVIEW RETURNED	20-Oct-2019

GENERAL COMMENTS	I thank the editor for allowing me to carry out this referee. The article is well structured and the method used is complete and punctual in all its parts. This study is very important for the scientific community and helps researchers and clinicians to evaluate stroke.
---

REVIEWER	Celine Gillebert and Hanne Huygelier KU Leuven, Belgium Please note that my PhD student, Hanne Huygelier, co-reviewed this manuscript.
REVIEW RETURNED	08-Nov-2019

GENERAL COMMENTS

I have read the manuscript, entitled “The Vision Screening Assessment (VISA) tool – diagnostic accuracy validation of a novel screening tool in detecting visual impairment among stroke survivors.” In this paper the authors validate a new tool to screen for visual impairments post-stroke. They recruited 222 stroke survivors in total of which 101 patients completed a pen-and-paper version and 100 patients completed an app version. Then, the authors test the agreement between their new VISA tool with an orthoptic visual assessment to validate their tool and they find relatively good agreement between the screening tool and the orthoptic visual assessment which is used as the reference.

I appreciate the effort to develop validated and standardized screening instruments tailored to stroke survivors, as this is indeed very important for clinical practice and I appreciate the effort in collecting a sufficiently large sample to validate the instrument. However, I do have many questions or concerns with the manuscript in its current form. I will elaborate on these aspects below.

1. In the first paragraph of the introduction the authors describe the importance of visual impairments post-stroke, but the concept of “visual impairments” is not clearly defined. Intact vision requires good functioning at many processing stages going from low (i.e., retinal) levels to higher levels of the visual processing hierarchy where information is integrated into meaningful percepts. I would advise the authors to frame the VISA tool more clearly within this more general framework of vision, and to discuss the literature coupled to this general framework. For instance, when stating that visual impairments occur in 65% of stroke survivors, I wonder to which types of visual impairments this number refers.

2. The authors do not discuss already existing tools to assess post-stroke visual impairments in their introduction (e.g. the Birmingham Object Recognition Battery, the Visual Object and Space Perception Battery, the Leuven Perceptual Organisation test, the Hooper Visual Organization Test, among others) and do not discuss the potential advantages of their new VISA tool in comparison to already existing tools. How does the VISA tool compare to already existing visual assessment instruments?

3. In the description of the methodology, the authors do not provide details about the different instruments and subtests of the VISA tool. This makes it difficult to understand the results and difficult to assess the clinical usefulness of the VISA tool. I outline the things that were unclear here below:

a. Case history: The authors do not describe how the case history could be obtained for patients that have aphasia. They also do not specify what kind of observations are used for the case history and in which way these observations are made. For instance, are these observations made by the examiner administering the VISA tool or are they asked from a caregiver who knows the patient? Do the observations involve formal items with a formal rating scale?

b. LogMAR visual acuity: Was the distance to the screen controlled or could patients freely adjust their own viewing distance? Do patients always read all lines of letters or do you break off the administration on the first line on which they make an error? If you administer the test twice (once at near distance, once at far distance) do you use the same order each time and the same letters? Do you administer this test if patients have expressive

aphasia or dysarthria and if so, do these problems affect the validity of the visual acuity assessment? How do post-stroke impairments such as neglect alexia, hemianopic alexia and visual extinction affect results on this assessment and how are these problems considered when interpreting test results? If the test procedure does not allow to adequately differentiate these problems in patients, this should be discussed as a limitation in the discussion.

c. Eye alignment and eye movements: Can you describe exactly what the patient is asked to do and what the examiner does to assess eye alignment and eye movements? What kind of cut-off value is used to decide whether the patient is impaired on these tests?

d. Visual field assessment: How do you assess visual field in the VISA print version? Does the examiner hold his hands in between him and the patient and does the examiner turn his/her hands slightly towards the patient? Do you use a fixed number of trials per visual quadrant? What does the patient need to report (e.g. which hand moves, how many fingers are shown) and how does the patient report this (e.g. by pointing, by a verbal answer)? What kind of cut-off value is used to decide whether the patient is impaired on these tests?

e. Assessment of visual inattention: Can you specify which type of line bisection and cancellation task you used and how they were administered? Can you specify how performance on the three tasks (i.e. line bisection, cancellation task and clock drawing) was summarized and how an impairment on these tests was determined? Details in how these tasks are administered, summarized and used to inform diagnosis can have a large impact on their validity (see for instance: Huygelier & Gillebert, 2018, 2019; McIntosh, 2017; McIntosh, Schindler, Birchall, & Milner, 2005; Toraldo, Romaniello, & Sommaruga, 2017) and are important to report so that their potential impact on the test's validity can be evaluated.

f. The kinetic visual field assessment of the VISA app: Can you provide details of what patients must do in this task and which type of visual information is presented to observers? What kind of cut-off value is used to decide whether the patient is impaired on these tests?

g. VISA app: Although the authors state that the print and app versions of the subtests are identical, it is unclear how the VISA app works exactly. Is it an app that clinicians use to provide input about the patient's performance while administering pen-and-paper based tests (except for the visual field test) or is it an app where all tests are presented in a digitalized format? On what type of technological system is the app used? Are aspects such as viewing distance, luminance, contrast controlled when using the VISA app?

4. The authors described the contents of the "orthoptic vision assessment" briefly, but it was unclear how this assessment differed from the VISA assessment. To be able to understand the results, it is necessary to understand how both assessments are similar or dissimilar to each other. For instant, what is meant by "assessment of visual perception"? It's also not clear whether the authors are referring to this assessment as the "specialist vision assessment" or whether these two are separate assessments. Please use consistent labelling of these assessments if they refer to the same assessment.

5. The authors describe that they assessed the chance-eliminated agreement using Kappa between the VISA results and results of the

orthoptic vision assessment and reported 95% confidence intervals. It is however unclear how the Kappa values and their confidence intervals were calculated (which formula or software package was used). Moreover, it's unclear whether the authors considered the impact of the base rate on the Kappa estimates (see for instance: Bruckner & Yoder, 2006). This seems highly relevant since the percentage of cases diagnosed with visual impairments on the orthoptic vision assessment (used as the reference) were strongly uneven (i.e., 85% was diagnosed with visual impairments) and likely varied across the different VISA subtests that are reported in Table 2, making the Kappa values incomparable if not corrected for base rates. Moreover, I am concerned that the strong imbalance between cases with and without visual impairments according to the reference may make the calculation of the 95% CI of Kappa invalid. Do the authors have a reference or proof that their method of calculating the 95% CI of Kappa is valid, given the unbalanced data?

6. The aim of the authors is to assess the validity of the VISA test and they assess the agreement of the VISA test with an orthoptic vision assessment. The authors should clarify why they chose to only assess the agreement with an orthoptic vision assessment and not other psychometric qualities of their instrument such as its internal consistency, construct validity, content validity, ecological validity. Perhaps these other psychometric aspects of test instruments that were not assessed should be highlighted as a limitation of the study.

7. The authors do not provide much information about the type of stroke patients that were assessed. Can authors provide information about the education level of patients, stroke etiology, lesion volume, lesion location, time of testing in days post-stroke, ...? Moreover, the samples of patients who completed the VISA Print versus VISA app were quite dissimilar in age and gender. Can the authors discuss how this may have affected their results? Moreover, the visual medical history of patients is not discussed; how many patients had a history of age-related visual problems such as cataract, glaucoma, ...? Please discuss how this medical history may affect your results.

8. In the results section the authors use qualitative descriptions to interpret the kappa values, sensitivity and specificity (e.g. "substantial agreement"), but do not provide an explanation on what this labelling was based.

9. In the data-analysis section the authors state that they will report 95% CIs for the sensitivity and specificity, while these are not reported in the Results.

10. Another minor note: if the presence of visual impairments is calculated across the different subtests, how is this done? Do you use the criterion of at least 1 impairment? How do you consider multiple comparisons for these calculations?

11. According to the orthoptic vision assessment, apparently 85% of patients had a visual impairment (I assume at least 1 visual impairment), but this differs from the prevalence of visual impairments reported in the introduction (i.e., 65%). Does this suggest that there was a bias in the orthoptic visual assessment to overestimate visual impairments or a bias in the sample favoring patients with visual impairments? Can the authors discuss how this may have affected their results?

VERSION 1 – AUTHOR RESPONSE

Reviewer: Thomas Meinel

Inclusion criteria: Severe cognitive impairment: defined as?

Statement defining what was meant by severe cognitive impairment in reference to the inclusion criteria has been added.

When was this done? Acute Stroke Care? Please report days after stroke as delirium, infections, blood pressure, ... can cause visual impairment.

The aim of the study was to assess whether the VISA tool was able to detect the presence of visual impairment when compared to a visual assessment conducted by a specialist i.e. orthoptist. The assessments were conducted within 24 hours of each other to minimise the chance of recovery between the assessments conducted. The assessments were done on the acute stroke unit as mentioned in the methods section and typically within one week of stroke onset with notes taken on the patient's medical status.

Discussion: "...3 days of stroke onset for at least half of stroke survivors and within 1 week of stroke onset for the majority" Suggest to do this test 4 weeks after stroke. Many motor symptoms of stroke are improving rapidly in the first days, so you might have picked up transient visual dysfunction, which might not have been relevant to the patient any more when he sees the specialist.

This was a UK study following UK national clinical guidelines for stroke. Thus screening was done as soon as possible after stroke onset. This reviewer raises an important point when suggesting delay of 4 weeks for vision screening. This was also a suggestion in the UK until research showed that the majority of those with visual impairment had limited or no recovery of their visual problem. Further there were often limited predictive factors for those who would show recovery. Therefore it is now recommended that all stroke survivors are seen quickly which avoids creating an unmet need for patients who are discharged home early with the consequence of having to live with an undiagnosed visual impairment without being given specialist support or treatment.

Convenience sample: How was randomization of which patients to screen exactly done? Are you sure, you did not exclude "hard to diagnose" patients, hence boosting the sens and spec of your tool?

No randomisation took place. Participants who were identified as being eligible were approached consecutively as a convenience sample for screening/orthoptic assessment. Each participant was tested by both the screener and orthoptist. This was a pragmatic study therefore the order of the screening and orthoptic assessment was varied, depending on which party was available to test the patient first. This has been clarified in the methods section.

We have stated the exclusion criteria; these are the only reasons patients were excluded from the study.

Introduction: when 65% of all patients have visual impairment and 60% of stroke survivors, then why is the visual impairment stroke-related? Misleading sentence.

65% is reporting the prevalence of visual impairment after stroke and 60% is reporting the incidence of new stroke-related onset visual impairment. This sentence has been amended to make this clearer.

"For the purpose of this study, vision screening was undertaken with VISA and screening was defined as the assessment of stroke survivors for the presence of reduced visual function against pre-set abnormality criteria." Which were they?

This criteria is outlined in the statistical methodology section. We have added a sign-post to this section for this information rather than repeating.

Did you publish your protocol?

No, the pilot study which followed a similar methodology was published in BMJ Open. This is referenced in the methods section.

“Visual field to confrontation”: Goldberg perimetry or finger?

More information has been added to clarify.

“The order of the VISA screening and orthoptic vision assessments varied”: by random?

More information has been added to clarify.

Screeener and orthoptist blinded to Imaging / Clinical Stroke Information?

This was a pragmatic clinical study. Both clinicians had access to the information they would normally have access to in their clinical roles. However the screener and orthoptist were blinded to the results of each other’s assessment.

Which patients VISA app, which VISA print? Randomized? 122 VISA print patients – 101 available = 21 missings: baseline differences? Hard to diagnose?

VISA print and VISA app were used consecutively – first the print version and then the app version once recruitment finished for the print version. Missing data was primarily from one site as the orthoptist did not carry out the visual inattention assessment routinely. This is outlined in the results section.

Agreement of whether to make a referral to specialist: was there a predefined cut-off or any visual impairment would lead to referral?

This is outlined in the statistical methodology and sample size section as follows: “A binary measure of the presence or absence of visual impairment (defined as reduced distance vision <0.2 , reduced near vision <0.3 (equivalent to N6), eye movement abnormality, visual field loss, visual perceptual abnormality).”

“The lowest level of agreement was produced for near visual acuity (0.236) and ocular motility (0.367), both with fair agreement.” Kappa <0.4 is usually considered poor agreement.

<https://www.sciencedirect.com/science/article/pii/S1556086415318876>

The interpretation we used for the kappa value along with the reference have been added to the statistical methodology and sample size section.

How were the abnormalities stroke related? Please report on stroke location (occipital cortex, brainstem, cerebellum) in order to get a feeling, how much of the visual impairment was actually stroke-related. Also suggest to do this screening in a matched population without stroke. I doubt a bit, how much of the pathologies were actually stroke-related. Specificity of 60%, imagine, what this would mean when VISA would be applied to all stroke patients in terms of logistics and finance. A referral of millions of patients to eye specialists for no reason. Please discuss this impact in your discussion.

We do not have the location of the stroke data available to us as this was not required for the study aim. The aim of the study was to assess whether the VISA tool was able to detect the presence of visual impairment when compared to a visual assessment conducted by a specialist i.e. orthoptist. For

the purposes of this study it was irrelevant whether the visual impairment was stroke related as we sought to detect pre-existent as well as new onset visual impairment. It is important to pick up patients with pre-existing visual impairment, especially in patients who may not be able to report this themselves as they may have lost the ability to use their established coping mechanisms. The incidence (the number of patients with new stroke related visual impairment) is reported elsewhere with full discussion of new versus prior visual problems and referenced within the paper (reference 4).

Table 3: please report frequencies of pathologic and normal results, not only agreement.

These figures have been added to table 1 and 3.

For which of these pathologies, there is actually an evidence-based treatment?

There are evidence-based treatments/management strategies available. Although provision of management information was not within the remit of this study, we have made reference to management in the discussion.

Reviewer 2: Giovanni Galeoto

We are grateful for this positive review.

Reviewer 3: Celine Gillebert and Hanne Huygelier

1. In the first paragraph of the introduction the authors describe the importance of visual impairments post-stroke, but the concept of “visual impairments” is not clearly defined. Intact vision requires good functioning at many processing stages going from low (i.e., retinal) levels to higher levels of the visual processing hierarchy where information is integrated into meaningful percepts. I would advise the authors to frame the VISA tool more clearly within this more general framework of vision, and to discuss the literature coupled to this general framework. For instance, when stating that visual impairments occur in 65% of stroke survivors, I wonder to which types of visual impairments this number refers.

The types of visual impairment have been added.

2. The authors do not discuss already existing tools to assess post-stroke visual impairments in their introduction (e.g. the Birmingham Object Recognition Battery, the Visual Object and Space Perception Battery, the Leuven Perceptual Organisation test, the Hooper Visual Organization Test, among others) and do not discuss the potential advantages of their new VISA tool in comparison to already existing tools. How does the VISA tool compare to already existing visual assessment instruments?

We are aware of these tools. However they are assessments for visual perception problems rather than the wider range of visual impairment including reduced visual acuity, eye movement problems, visual field loss, visual inattention and visual perception problems. A systematic review has shown that such a tool does not exist and also highlighted the limitations of the checklists for this population as outlined in the discussion.

3. In the description of the methodology, the authors do not provide details about the different instruments and subtests of the VISA tool. This makes it difficult to understand the results and difficult to assess the clinical usefulness of the VISA tool. I outline the things that were unclear here below:

a. Case history: The authors do not describe how the case history could be obtained for patients that have aphasia. They also do not specify what kind of observations are used for the case history and in which way these observations are made. For instance, are these observations made by the examiner administering the VISA tool or are they asked from a caregiver who knows the patient? Do the observations involve formal items with a formal rating scale?

More information has been added to answer these queries.

b. LogMAR visual acuity: Was the distance to the screen controlled or could patients freely adjust their own viewing distance? Do patients always read all lines of letters or do you break off the administration on the first line on which they make an error? If you administer the test twice (once at near distance, once at far distance) do you use the same order each time and the same letters? Do you administer this test if patients have expressive aphasia or dysarthria and if so, do these problems affect the validity of the visual acuity assessment? How do post-stroke impairments such as neglect alexia, hemianopic alexia and visual extinction affect results on this assessment and how are these problems considered when interpreting test results? If the test procedure does not allow to adequately differentiate these problems in patients, this should be discussed as a limitation in the discussion.

Distances for near and distance tests have been added. The alternatives offered for patients who are unable to name letters have been added.

c. Eye alignment and eye movements: Can you describe exactly what the patient is asked to do and what the examiner does to assess eye alignment and eye movements? What kind of cut-off value is used to decide whether the patient is impaired on these tests?

More information has been added to answer these queries.

d. Visual field assessment: How do you assess visual field in the VISA print version? Does the examiner hold his hands in between him and the patient and does the examiner turn his/her hands slightly towards the patient? Do you use a fixed number of trials per visual quadrant? What does the patient need to report (e.g. which hand moves, how many fingers are shown) and how does the patient report this (e.g. by pointing, by a verbal answer)? What kind of cut-off value is used to decide whether the patient is impaired on these tests?

Information on visual field test methods for both VISA print and app has been added.

e. Assessment of visual inattention: Can you specify which type of line bisection and cancellation task you used and how they were administered? Can you specify how performance on the three tasks (i.e. line bisection, cancellation task and clock drawing) was summarized and how an impairment on these tests was determined? Details in how these tasks are administered, summarized and used to inform diagnosis can have a large impact on their validity (see for instance: Huygelier & Gillebert, 2018, 2019; McIntosh, 2017; McIntosh, Schindler, Birchall, & Milner, 2005; Toraldo, Romaniello, & Sommaruga, 2017) and are important to report so that their potential impact on the test's validity can be evaluated.

Further information has been added to the methods section.

f. The kinetic visual field assessment of the VISA app: Can you provide details of what patients must do in this task and which type of visual information is presented to observers? What kind of cut-off value is used to decide whether the patient is impaired on these tests?

More information has been added to answer these queries. Referral guidelines are outlined in the methods.

g. VISA app: Although the authors state that the print and app versions of the subtests are identical, it is unclear how the VISA app works exactly. Is it an app that clinicians use to provide input about the patient's performance while administering pen-and-paper based tests (except for the visual field test) or is it an app where all tests are presented in a digitalized format? On what type of technological system is the app used? Are aspects such as viewing distance, luminance, contrast controlled when using the VISA app?

The assessments in VISA app are identical to VISA print (except visual fields as indicated). The clinician and patient use a tablet device (e.g. iPad) on which the VISA app was downloaded. Each section is used and the results captured and reported as a pdf record.

4. The authors described the contents of the "orthoptic vision assessment" briefly, but it was unclear how this assessment differed from the VISA assessment. To be able to understand the results, it is necessary to understand how both assessments are similar or dissimilar to each other. For instance, what is meant by "assessment of visual perception"? It's also not clear whether the authors are referring to this assessment as the "specialist vision assessment" or whether these two are separate assessments. Please use consistent labelling of these assessments if they refer to the same assessment.

The labelling of assessment has been made consistent and is now orthoptic vision assessment. However during the discussion it is referred to as specialist vision assessment, as these assessments could be conducted by other eye-specialists.

The orthoptic assessment has similar elements to the VISA tool, with the main difference being that it is conducted by an orthoptist with specific expertise in neurological/stroke visual assessments.

Visual perception was a typo error in this section and has been corrected.

5. The authors describe that they assessed the chance-eliminated agreement using Kappa between the VISA results and results of the orthoptic vision assessment and reported 95% confidence intervals. It is however unclear how the Kappa values and their confidence intervals were calculated (which formula or software package was used). Moreover, it's unclear whether the authors considered the impact of the base rate on the Kappa estimates (see for instance: Bruckner & Yoder, 2006). This seems highly relevant since the percentage of cases diagnosed with visual impairments on the orthoptic vision assessment (used as the reference) were strongly uneven (i.e., 85% was diagnosed with visual impairments) and likely varied across the different VISA subtests that are reported in Table 2, making the Kappa values incomparable if not corrected for base rates. Moreover, I am concerned that the strong imbalance between cases with and without visual impairments according to the reference may make the calculation of the 95% CI of Kappa invalid. Do the authors have a reference or proof that their method of calculating the 95% CI of Kappa is valid, given the unbalanced data?

The statistical software used has been added. We thank the reviewer for raising this concern and requested a full statistical review of our analysis – this statistician has been added to the list of authors. We ran additional analysis using the method outlined by Bruckner & Yoder and, as this did not change the interpretation of the results we have not included this analysis but have added to the discussion section regarding the interpretation of Kappa values and consideration of the base rates.

6. The aim of the authors is to assess the validity of the VISA test and they assess the agreement of the VISA test with an orthoptic vision assessment. The authors should clarify why they chose to only assess the agreement with an orthoptic vision assessment and not other psychometric qualities of their instrument such as its internal consistency, construct validity, content validity, ecological validity. Perhaps these other psychometric aspects of test instruments that were not assessed should be highlighted as a limitation of the study.

We have added further discussion of these points in the discussion section. We had originally made reference to process evaluation but with insufficient detail.

7. The authors do not provide much information about the type of stroke patients that were assessed. Can authors provide information about the education level of patients, stroke etiology, lesion volume, lesion location, time of testing in days post-stroke, ...?

We did not gather information on education level or lesions site/size as this information was not always available but was also not necessary to the primary aim of this study.

Moreover, the samples of patients who completed the VISA Print versus VISA app were quite dissimilar in age and gender. Can the authors discuss how this may have affected their results? Moreover, the visual medical history of patients is not discussed; how many patients had a history of age-related visual problems such as cataract, glaucoma, ...? Please discuss how this medical history may affect your results.

We have added discussion of missing demographic, stroke and ocular history information.

8. In the results section the authors use qualitative descriptions to interpret the kappa values, sensitivity and specificity (e.g. "substantial agreement"), but do not provide an explanation on what this labelling was based.

The interpretation of Kappa used and the reference have been added to the statistical methodology section.

9. In the data-analysis section the authors state that they will report 95% CIs for the sensitivity and specificity, while these are not reported in the Results.

95% CIs for sensitivity and specificity are reported in tables 1 and 3.

10. Another minor note: if the presence of visual impairments is calculated across the different subtests, how is this done? Do you use the criterion of at least 1 impairment? How do you consider multiple comparisons for these calculations?

This has been clarified in the statistical methodology section.

11. According to the orthoptic vision assessment, apparently 85% of patients had a visual impairment (I assume at least 1 visual impairment), but this differs from the prevalence of visual impairments reported in the introduction (i.e., 65%). Does this suggest that there was a bias in the orthoptic visual assessment to overestimate visual impairments or a bias in the sample favoring patients with visual impairments? Can the authors discuss how this may have affected their results?

The aim of the study was to assess whether the VISA tool was able to detect the presence of visual impairment when compared to a visual assessment conducted by an orthoptist. This was a pragmatic study with a convenience sample and as such therapists were more likely to conduct this assessment if they were suspicious of a visual impairment. Further discussion has been added to the limitations section.

VERSION 2 – REVIEW

REVIEWER	Thomas Meinel Inselspital
REVIEW RETURNED	21-Feb-2020

GENERAL COMMENTS	I think this tool is certainly interesting, but without information on stroke location, and clinical parameters I do not see why this study was done in stroke patients after all. Although the authors state, that their aim was to assess all visual problems (also unrelated to stroke). Also without information on the patient characteristics, the statement that VISA is acceptable for screening simply is not justified, because you do not know which patients you should actually approach with this tool.
---